# Detecting Origin Attribution for Text-to-Image Diffusion Models in RGB and Beyond

## Abstract

1  Modern text-to-image (T2I) diffusion models can generate images with remarkable
2  realism and creativity. These advancements have sparked research in fake image
3  detection and attribution, yet prior studies have not fully explored the practical and
4  scientific dimensions of this task. In this work, we not only attribute images to 12
5  state-of-the-art T2I generators but also investigate what inference stage hyperpa-
6  rameters are discernible. We further examine what visual traces are leveraged in
7  origin attribution by perturbing high-frequency details and employing mid-level
8  representations of image style and structure. Notably, altering high-frequency infor-
9  mation causes only slight reductions in accuracy, and training an attributor on style
10  representations outperforms training on RGB images. Our analyses underscore that
11  fake images are detectable and attributable at various levels of visual granularity.

## 1  Introduction

13  Recent text-to-image (T2I) diffusion models [4, 32, 41, 43, 45, 46, 49, 51] have markedly transformed
14  image generation, enabling the creation of highly realistic and imaginative visual content directly
15  from textual descriptions. However, this progress introduces significant challenges in discerning
16  real images from AI-generated images and accurately identifying their origins. Addressing these
17  challenges is vital for preserving the integrity of visual content across digital platforms.

18  Previous studies [2, 5, 7, 26, 56, 61, 66] have focused on differentiating AI-generated images from
19  real ones, with some research attributing images to their source generators, notably in GAN variants
20  [6, 21, 35, 63] and diffusion models [11, 22, 54]. Yet, these investigations have largely been conducted
21  using generative models that may not reflect the latest advancements, and they have not fully explored
22  the practical and scientific dimensions of this task, which we aim to further examine.

## 2  Dataset Generation

24  In this work, we detect origin attributions for modern text-to-image (T2I) models, while also investi-
25  gating the extent to which traces are detectable across generators and inference stage controls. To
26  achieve this, we generate images using a variety of T2I models and text prompts to ensure diversity.
27  Additionally, we maintain a consistent generator and adjust inference time hyperparameters.

### 2.1  Images from Diverse Generators and Prompts

29  As depicted in Fig. 1, we employed 12 modern, open-source T2I models for image generation: SD
30  1.5 [49], SD 2.0 [49], SDXL [43], SDXL Turbo [51], Latent Consistency Model (LCM) [32], Stable
31  Cascade [41], Kandinsky 2 [46], DALL-E 2 [45], DALL-E 3 [4], and Midjourney versions 5.2 and
32  6 [38]. To generate images, we use the OpenAI API for DALL-E 2 and 3, an automation bot for
33  Midjourney 5.2 and 6, and the Hugging Face diffusers repository [60] for the remaining models. To
34  gather a broad spectrum of text prompts, we leveraged around 5,000 captions from MS-COCO [30].

Figure 1: A depiction of images generated for our dataset using 12 different T2I generators.

### 2.2  Images from Varying Hyperparameters During Inference Stage

36  We expand our focus beyond identifying the source generators based on their architectures, to a
37  deeper analysis of the critical yet subtle choices made during the inference stage that greatly impact

the generated outputs. Initially, we investigate the possibility of identifying certain checkpoints within the same architecture, specifically Stable Diffusion (SD) [49], based on different training iterations. Next, we question whether the generated images can reveal which scheduler [23, 27, 57, 65] was employed during the inference phase for the same generator. Furthermore, drawing inspiration from studies indicating that different seed numbers in GAN-generated images can be detected [64], we apply this concept to diffusion models to determine if the seed is detectable based on the images. Finally, we conduct experiments with diffusion steps ranging from 5 to 50 in increments of 5 to investigate whether the number of sampling steps leaves detectable traces in the images. Selected samples of images generated under different hyperparameter adjustments are presented in Fig. 2.

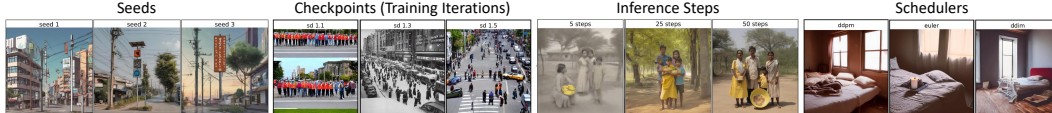

Figure 2: We show the diversity in generated images influenced by varying hyperparameters, such as checkpoints of the same architecture, schedulers, initialization seeds, and number of inference steps.

## 3 Detecting Origin Attribution in RGB

In this section, we benchmark origin attribution performance across 12 modern text-to-image generators, examining the impact of various architectures and training sizes on task performance. We then analyze the detectability of traces for various hyperparameter adjustments during inference time.

### 3.1 Training Origin Attributors

**Problem Setup and Model Performance.** Our study merges the tasks of discerning "AI-generated vs. Real Images" and attributing images to their sources. This is achieved by including real images in our dataset and treating them as an additional 'generator'. Concerning the architecture of the origin attributor, which functions as an image classifier, prior work [12, 39] showed that a straightforward linear probe or nearest neighbor search applied to a large pretrained model like CLIP [44] can effectively differentiate AI-generated images from real ones. Inspired by these findings, we employ three network architectures to tackle our attribution task: an EfficientFormer [28] trained from scratch, a CLIP [44] backbone connected with a linear probe and MLP, and DINOv2 [40] with a similar configuration. We also analyze the impact of incorporating text prompts as inputs similar to Sha et al. [54], providing slight yet consistent improvements across all architectures, as shown in Tab. 1.

|           | E.F. (scratch) | CLIP + LP | CLIP + MLP | DINOv2 + LP | DINOv2 + MLP |
|-----------|----------------|-----------|------------|-------------|--------------|
| w/o text  | 90.03%         | 70.15%    | 73.09%     | 67.68%      | 71.33%       |
| w/ text   | 90.96%         | 71.44%    | 74.19%     | 69.44%      | 73.08%       |

Table 1: The 13-way classification accuracy of various architectures for origin attribution performed across 12 generators and a set of real images, with each class containing an equal number of images. The probability of randomly guessing the correct source is $\frac{1}{13}$, which gives a **7.69%** accuracy. "E.F." refers to EfficientFormer trained from scratch. The first and second rows in the results table indicate classifiers trained without and with text prompts, respectively.

**Classifier Performance Across Generators.** As illustrated in Fig. 3, there is a noticeable challenge in differentiating generators from the same family, with notable pairs including "SD 1.5 vs. SD 2.0," "Midjourney 5.2 vs. Midjourney 6," and "LCM (2 steps) vs. LCM (4 steps)." While Midjourney's architecture remains undisclosed to the public, it is reasonable to infer that versions 5.2 and 6 likely share a similar underlying architecture from our analysis. Interestingly, DALL-E 3 presents more confusion when compared to Midjourney versions 5.2 / 6, rather than with DALL-E 2. We attribute this finding to the significant architectural differences: DALL-E 2 incorporates pixel diffusion in its decoder stage, whereas DALL-E 3 employs multi-stage latent diffusion alongside a distinct one-step VAE decoder, similar to [49], leading to divergent generative characteristics. Finally, we demonstrate that the accuracy of the attributor consistently improves with an increase in the number of training images, as shown on the right side of Fig. 3. However, due to budget constraints, fully exploring the dataset expansion up to the saturation point is deferred to future research endeavors.

### 3.2 Analyzing the Detectability of Hyperparameter Variations

T2I generators often have several hyperparameters at the inference stage that impact the generated image quality, and a natural question that arises is whether images produced using different hyperparameters are distinguishable. To investigate this, we target four hyperparameter choices for Stable Diffusion [49]: model checkpoint, scheduler type, number of sampling steps, and initialization seed.

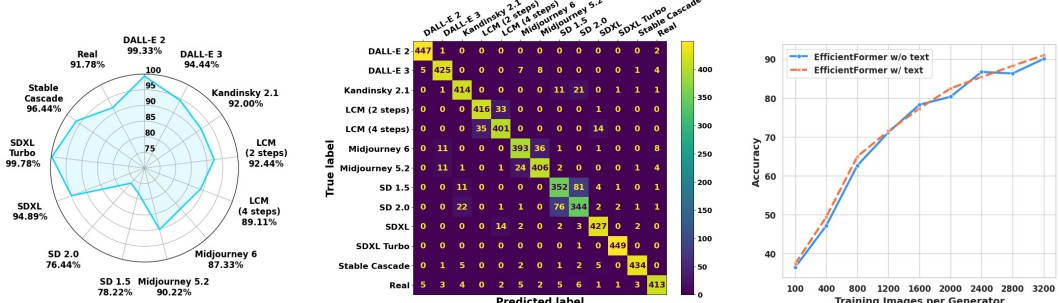

Figure 3: **Left/Middle:** Accuracy and confusion matrix of EfficientFormer trained with prompts, which had the best accuracy. **Right:** Accuracy of EfficientFormer as we vary the number of images.

Specifically, we compared Stable Diffusion checkpoints 1.1 to 1.5, each of which is trained using a different number of iterations on LAION [52]. We then examined the detectability of images generated using eight schedulers: DDIM [57], DDPM [23], Euler [27], Euler with ancestral sampling [27], KDPM 2 [27], LMS [27], PNDM [27], and UniPC [65]. Additionally, we generated images using SD 2.0 and SDXL for ten different sampling steps ranging from 5 to 50, and ten different seeds ranging from 1 to 10. For each hyperparameter, we train a separate EfficientFormer [28] to classify the generated images. As shown in Tab. 2, all six classifiers detect the hyperparameter choice better than random chance. Interestingly, detecting the initialization seed achieves nearly 100% accuracy, aligning with work by Yu et al. [63] that found different seeds lead to attributable GAN fingerprints. Moreover, based on the confusion matrix for different sampling steps using SDXL in Fig. 4, we see that images generated using fewer steps are more detectable than those generated using more steps, likely because fewer steps noticeably degrades the generation quality.

Figure 4: Confusion matrices for hyperparameter variations. We observe that images generated with fewer SDXL sampling steps are more detectable, likely due to visible degradation in image quality.

|  | Checkpoints | Schedulers | Sampling Steps | Seeds |
|---|---|---|---|---|
| Random | 20% | 12.5% | 10% / 10% | 10% / 10% |
| Accuracy | 30.21% | 20.18% | 25.96% / 56.64% | 98.80% / 99.94% |

Table 2: Comparison of accuracy for detecting hyperparameter values based on generated images. For the 'Sampling Steps' and 'Seeds' trials, we trained and evaluated on images from SD 2.0 and SDXL. Accuracies are written as *SD 2.0 / SDXL*. Notably, the 'Seeds' trial has near perfect performance.

# 4 Detecting Origin Attribution Beyond RGB

Previous studies have suggested that an origin attributor may leverage middle-to-high frequency information to differentiate images. However, it remains unclear what constitutes "middle-to-high frequency information" and to what extent the network can identify detectable traces in the images. Thus, we present an extensive empirical study on the impact of incrementally eliminating visual details at various levels of granularity on origin attribution performance.

**High-Frequency Perturbations.** Prior research [3, 5, 10, 13, 15–17, 34, 48, 59] has identified that generators leave unique fingerprints in the high-frequency domain, allowing attributors to learn these high-frequency details effectively. As an initial step, we investigate the effects of introducing high-frequency perturbations to images on the attributor's performance, which aims to enforce the classifier to learn beyond high-frequency details. For simplicity, we train a separate EfficientFormer [28] on each set of perturbed images. Figure 5 illustrates our observations under four types of perturbation: Gaussian blur, bilateral filtering, adding Gaussian noise, and SDEdit [36]. We note that these perturbations result in a modest decrease in classification accuracy. Specifically for SDEdit, the high-frequency traits of SDXL are embedded into every image, regardless of their source generators, by undergoing processing via the encoder, diffusion UNet, and decoder of SDXL [43]. Remarkably,

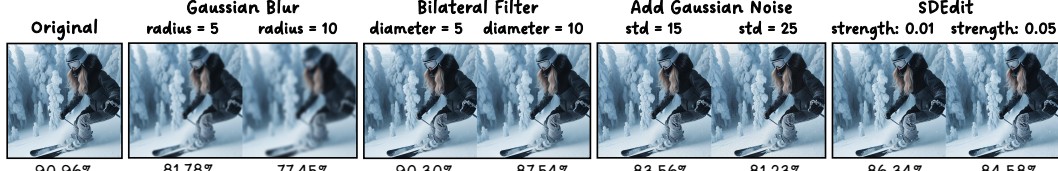

Figure 5: We present a generated image before and after perturbing its high-frequency details. We trained EfficientFormer on images after each high-frequency perturbation and observed a mild decline in the respective test accuracy, as shown beside the images.

this process led to only a minor reduction in accuracy, suggesting a robustness in the attributor's ability to identify generator-specific fingerprints despite high-frequency modifications.

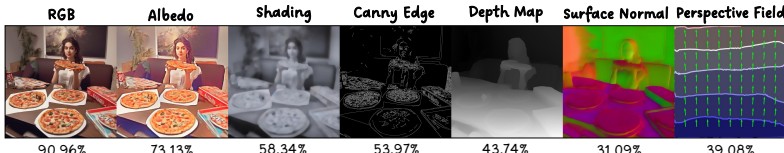

Figure 6: We present an RGB image and its mid-level representations. We trained EfficientFormer on each representation and show the test accuracy below each image. Note that random chance is 7.69%.

**Middle-Level Representations.** High-frequency perturbations result in only minor performance degradation, suggesting that the detectable traces left by different generators might also reside within the mid-frequency domain. To study the presence of these detectable traces, we convert the images into various mid-level representations—'Albedo' [14], 'Shading' [14], 'Canny Edge', 'Depth Map' [62], 'Surface Normal' [14], and 'Perspective Fields' [24]—utilizing readily available models. This approach aims to uncover the extent to which these mid-level frequencies carry generator-specific information that can be used for attribution. We proceed by training a distinct EfficientFormer [28] for each mid-level representation, and we show their classification accuracies in Fig. 6. Notably, although the overall accuracy for the attributors trained on Canny Edge, Depth Map, and Perspective Field images is not high, they demonstrate remarkable performance at discerning real vs. fake images in Fig. 8 of the supplemental. This finding aligns with work by Sarkar et al. [50] suggesting that generative models often fail to generate accurate geometry.

**Image Style Representations.** Furthermore, it's common to observe perceptible style differences among outputs of image generators. For instance, Midjourney [38] often produces images with a 'cinematic' quality, while DALL-E [4, 45] tends to create images with overly smooth textures and cartoonish appearances, as in Fig. 1. This observation leads to a pertinent question: if we train an attributor on only stylistic representations of images, can we still identify source generators?

To capture the style representation of images, we adhere to methods from style transfer literature [19, 25], employing a pretrained VGG network [55] to extract features across multiple layers. Next, we compute the Gram matrix [18] for each network layer. If we denote the feature at a specific layer as $F \in \mathbb{R}^{H \times W \times N}$, then the Gram matrix $G \in \mathbb{R}^{N \times N}$ is the cosine similarity between each channel in the feature representation. This process distills the style of images, providing a unique fingerprint for each generator's output. We reshape and concatenate the Gram matrices extracted from multiple layers, and then train EfficientFormer [28] using these aggregated feature vectors.

Remarkably, the origin attributor achieves an accuracy of **92.80%** when trained on style representations, surpassing the performance of the attributor trained on original RGB images by **1.84%**. The superior accuracy from this style-based attributor highlights the importance of stylistic features, such as texture and color patterns, in discerning generators more effectively than the direct visual content. This insight not only advances our understanding of origin attribution techniques but also emphasizes the potential of leveraging stylistic elements for more nuanced AI recognition and analysis tasks.

## 5 Conclusion

In this study, we present in-depth analyses on detecting and attributing images generated by modern text-to-image (T2I) diffusion models. Our origin attributors, trained to recognize outputs from 12 T2I diffusion models along with a class of real images, reached an impressive accuracy of over 90% that significantly surpasses random chance. Additionally, our investigations into the challenge of distinguishing generators within the same family and the detectability of hyperparameter choices at inference time provide comprehensive insights. Going beyond mere RGB analysis, we introduce a new framework for identifying detectable traces across levels of visual detail, offering profound insights into the underlying mechanics of origin attribution. These analyses provide fresh perspectives on image forensics aimed at alleviating the threat of synthetic images.

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

## A  Human Performance

In computer vision and machine learning, human performance is typically seen as the benchmark for AI models. However, in the case of origin attribution, the scenario reverses—AI significantly outperforms humans. This is highlighted by an experiment conducted by one of our co-authors, who has extensive experience with AI-generated images. Tasked with attributing 650 images to their source generators, the co-author only achieved a **37.23%** accuracy. Although better than the 7.69% random chance level, this figure is markedly inferior to the over 90% accuracy of our top AI classifier. This outcome underlines the exceptional challenge of origin attribution, where even well-informed individuals struggle. It also shows the need for AI in assisting humans with tasks beyond their natural proficiency, emphasizing AI's potential to enhance human performance in specialized domains.

From the perspective of the human evaluator, differentiating between certain AI image generators and others can be nuanced yet discernible. The Latent Consistency Models (LCM) [32] at 2 and 4 steps are notable for their occasional oversmooth artifacts, a result of undersampling, making them easier to identify compared to other models. DALL-E 3 [4] is distinguished by its tendency to produce surreal, cartoonish images, though these often exhibit repetitive patterns. DALL-E 2 [45], on the other hand, is characterized by a unique 'sharp' visual artifact, likely a consequence of its pixel diffusion process in the decoder, setting it apart from other models. Midjourney versions 5.2 and 6 [38] typically deliver the highest quality images, sometimes with a distinctive cinematic style.

Real images, however, are more straightforward to identify. One can often look at detailed object regions—like hands and text—where AI-generated images tend to falter. The naturalistic photo style of real images serves as a key differentiation factor from AI-generated content. Other generators, such as SD 1.5 [49], SD 2.0 [49], SDXL [43], SDXL Turbo [51], Kandinsky 2.1 [46], and Stable Cascade [41], present a greater challenge for humans to distinguish due to the subtlety of their differences.

## B  Elaboration on Results in the Main Paper

**Training Origin Attributors.** In Sec. 3.1, we trained an EfficientFormer [28] trained from scratch, a CLIP [44] backbone connected with a linear probe and MLP, and DINOv2 [40] with a similar configuration as our origin attributors. Figure 7 and Table 3 showcase the confusion matrices and evaluation metrics for these attributors.

|  | E.F. (scratch) | CLIP+LP | CLIP+MLP | DINOv2+LP | DINOv2+MLP |
|---|---|---|---|---|---|
| Accuracy | 90.03 / 90.96 | 70.15 / 71.44 | 73.09 / 74.19 | 67.68 / 69.44 | 71.33 / 73.08 |
| Precision | 90.07 / 90.98 | 69.95 / 71.30 | 73.13 / 74.12 | 67.36 / 69.09 | 71.20 / 72.91 |
| Recall | 90.03 / 90.96 | 70.15 / 71.44 | 73.09 / 74.19 | 67.68 / 69.44 | 71.33 / 73.08 |
| F1 | 90.04 / 90.96 | 70.00 / 71.25 | 73.07 / 74.12 | 67.45 / 69.17 | 71.23 / 72.93 |

Table 3: Additional quantitative evaluation of image attributors for 13-way classification, consisting of 12 generators and a set of real images. The values (percentages) represent training each attributor *Without / With* text prompts.

**Takeaways from High-Frequency Perturbations.** Prior works have predominantly claimed that classifiers in tasks like 'real vs. fake' and origin attribution primarily learn from discriminative information in the high-frequency domain. While we concur that high-frequency details can be crucial for discrimination, our work has demonstrated that even when these details are altered, the classifier can still identify highly discriminative features and attain decent accuracy. Our finding does not contradict earlier claims, but rather suggests a shift in perspective, showing that reliance only on high-frequency details may not be necessary.

**Middle-Level Representations.** In Sec. 5, we trained a distinct EfficientFormer [28] on six mid-level visual representations of the AI-generated images, and Fig. 8 showcases their confusion matrices.

## C  Data and Implementation Details

**Image Generation.** We employed 12 T2I diffusion models to generate RGB images without watermarks, and the generated image sizes are as follows:

- **512 × 512:** Kandinsky 2.1, SD 1.1, SD 1.2, SD 1.3, SD 1.4, SD 1.5, SD 2.0, SDXL Turbo
- **1024 × 1024:** DALL-E 2, DALL-E 3, LCM (2 steps), LCM (4 steps), Midjourney 5.2, Midjourney 6, SDXL, Stable Cascade

We also use 5000 real images from the MS-COCO [29] 2017 validation set.

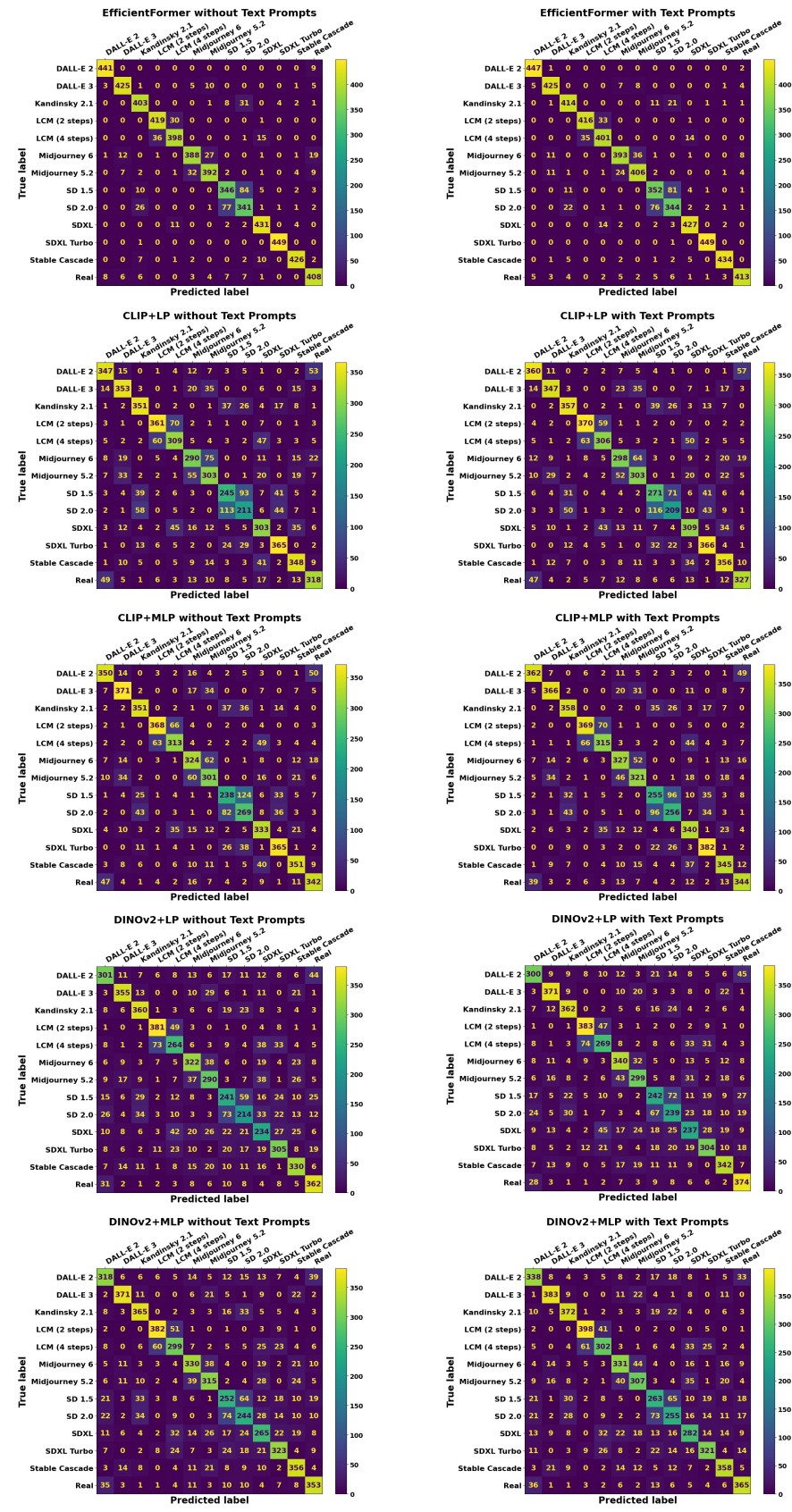

Figure 7: Confusion matrices for origin attributors in Sec. 3.1. The backbone for the CLIP and DINOv2 models is frozen.

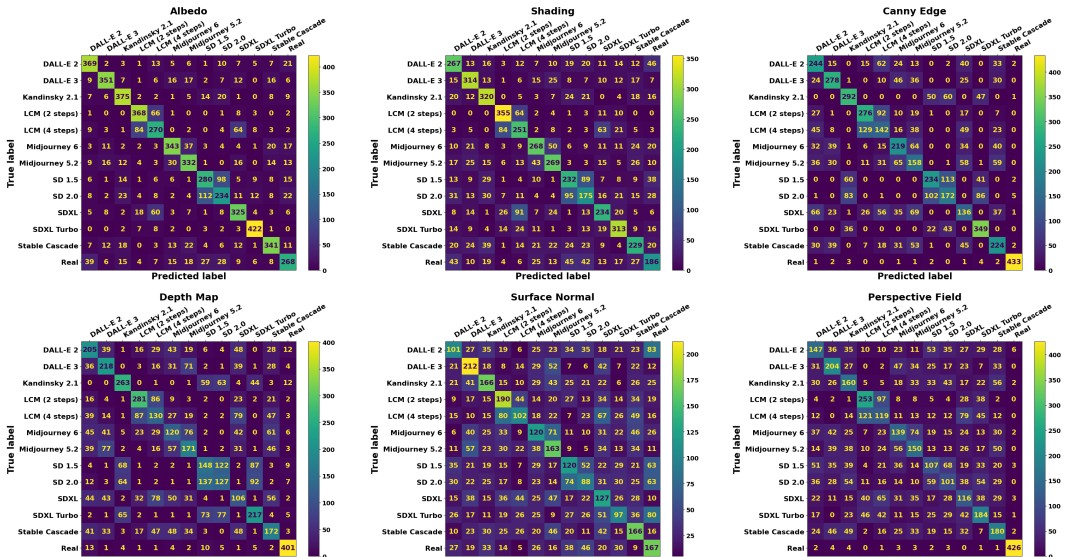

Figure 8: Confusion matrices for origin attributors trained on mid-level representations. Remarkably, attributors trained on "Canny Edge," "Depth Map," and "Perspective Field" images are significantly better at detecting real images than synthetic images.

**More Visualizations of Hyperparameter Variations.** As an extension of Fig. 2 in the main paper, we show more image generations with hyperparameter variations in Fig. 9 in the supplemental.

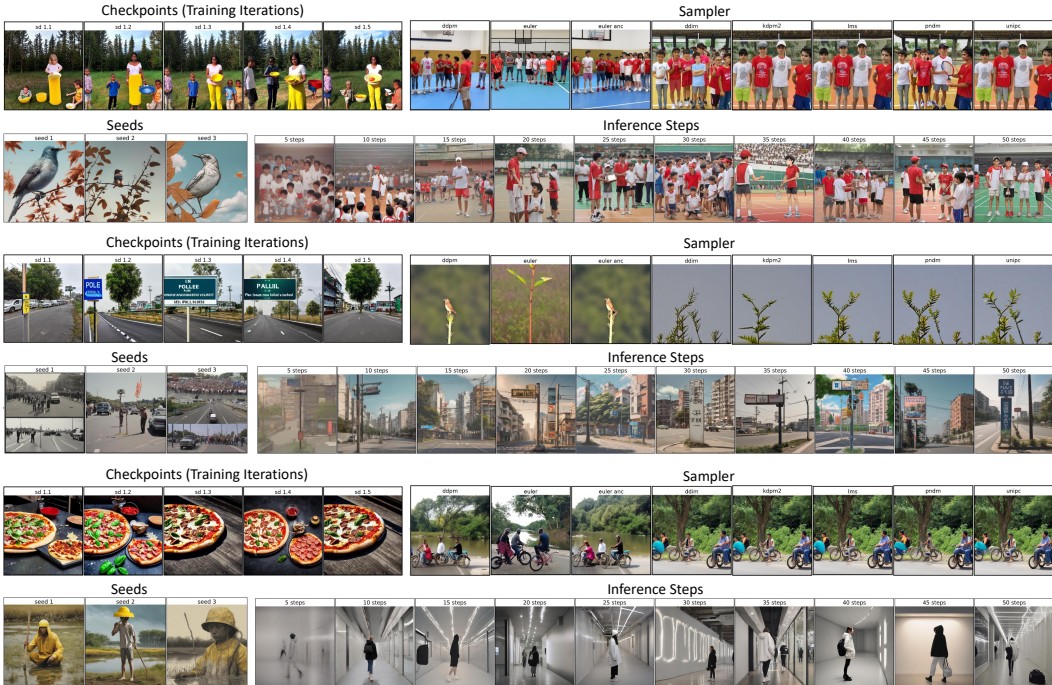

Figure 9: More examples showcasing the diversity in generated images influenced by varying hyperparameters: different model checkpoints within the same architecture, diverse scheduling algorithms, varied initialization seeds, and a range of inference steps.

**Training Data.** For Sec. 3.1 and 4 in the main paper, we view origin attribution as a 13-way classification task with 12 text-to-image diffusion models and a set of real images. It's important to note that we use 3200 training, 450 validation, and 450 testing images per class.

For Sec. 3.2, we analyze four hyperparameters: Stable Diffusion checkpoint, scheduler type, number of sampling steps, and initialization seed. When training classifiers for SD checkpoints, schedulers, and sampling steps, we use 20000 training, 2500 validation, and 2500 testing images per class. For seeds, we use 3200 training, 450 validation, and 450 testing images per class.

**Data Augmentation.** During training, we resize each image to have a shorter edge of size 224 using bicubic interpolation, center crop the image to size $224 \times 224$, and finally randomly horizontal flip the image with probability $0.5$. During validation and testing, we only resize and center crop images.

**Origin Attributors.** We selected three network architectures for the origin attribution task, and we use the code implementation from MMPretrain [9]. Our primary architecture is EfficientFormer-L3 [28] trained from scratch because it is a lightweight transformer. Moreover, we employ a pretrained, frozen transformer backbone attached to a linear probe (LP) or multilayer perceptron (MLP). The backbone is either CLIP ViT-B/16 [44] or DINOv2 ViT-L/14 [40], and the MLP consists of three linear layers with sigmoid activation and hidden dimension 256. For the linear probe and MLP classifier heads, there are 768 channels in the input feature map for CLIP+LP and CLIP+MLP, and 1024 channels for DINOv2+LP and DINOv2+MLP.

To train origin attributors with text prompts, we compute text embeddings using a pretrained CLIP [44] text encoder. Then, we concatenate image embeddings from the backbone with text embeddings as input to the classifier head.

For all origin attributors, we set a batch size of 128 and train for 2000 epochs. We use the checkpoint with the best validation accuracy. Additionally, we utilize the AdamW optimizer [31] with learning rate 0.0002 and weight decay 0.05. The learning rate scheduler has a linear warm-up period of 20 epochs, followed by a cosine annealing schedule with a minimum learning rate of 0.00001.

**Perspective Fields.** We use the code implementation from [24]. Each input to the attributor trained on Perspective Fields has a size of $640 \times 640 \times 3$. The first $640 \times 640$ channel contains latitude values, and the next two $640 \times 640$ channels contain gravity values. We adapt the code from [24] to visualize the Perspective Field on a black image in Fig. 6 of the main paper.

**How Gram Matrix Relates to Image Style.** Gatys et al. [18] characterize the texture of an image by computing correlations between feature channels in each layer of a convolutional neural network. These correlations are given by the Gram matrix, which is the inner product of vectorized feature maps. Extending their method to image style, Gatys et al. [19] incorporate feature correlations, i.e. Gram matrices, from multiple layers of the network to obtain a multi-scale representation of the image that extracts texture details without the global arrangement. Intuitively, employing different layers of the network leads to style representations at varying scales because features capture more complex information in later network layers. Thus, we aggregate Gram matrices from three layers of a pretrained VGG network to train our origin attributor on image style representations.

**Adapting to New Text-to-Image Diffusion Models.** Our work provides a seamless integration pathway for new generative models. For instance, to incorporate a new generator such as SD 2.0, one would simply generate approximately 5,000 images, add them to the existing training dataset, and retrain the models. This process typically requires around three days using a single RTX 4090 GPU. We intend to continually update our origin attributor to include popular new open-source generators. Moreover, should there be a model not yet incorporated, anyone could replicate this integration process independently, as we plan to release all related code and datasets to the community.

# D  Additional Experiments

## D.1  Detectability of Post-Editing Enhancements

A common workflow for utilizing AI-generated images involves users identifying unwanted artifacts or distracting areas within these images. They often import these images into additional models or software for further editing and refinement, such as using SDXL Inpainting [43] or Photoshop Generative Fill (Ps GenFill) [42] to enhance local regions. Many T2I applications are limited to relatively low resolutions, typically around 1K, or produce images with smooth/blurry texture. Hence, some professionals upscale or refine the details of generated images using advanced tools, such as Magnific AI [33]. This practice leads to a pertinent question: Is it possible to still detect the original source generator after the images have undergone further modifications using a variety of software or other AI models? For instance, an image initially created by Midjourney 6 [38] could subsequently be edited with SDXL Inpainting, Photoshop GenFill, or Magnific AI, as illustrated in Fig. 10.

To simulate typical user edits, we generated free-form masks across three size categories—small (0 to 15%), medium (15 to 30%), and large (30 to 60%)—reflecting the common range of edits applied to images. These masks were applied to the entire test set for pixel regeneration using SDXL Inpainting [43] and Ps GenFill [42]. We then assessed the best performing image attributor in Tab.

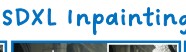
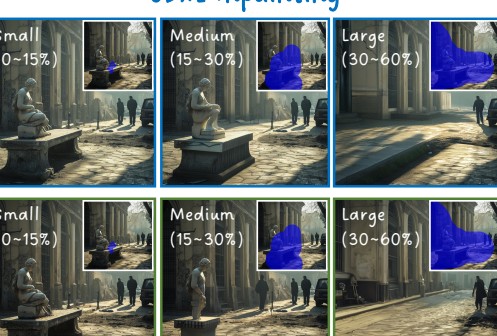
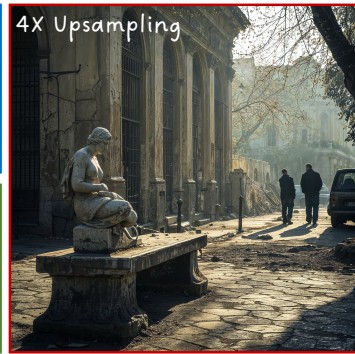
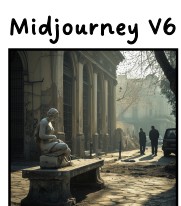

Figure 10: **Left**: Original image generated by Midjourney 6. **Middle**: Local modifications utilizing SDXL inpainting and Photoshop Generative Fill across three masks with small, medium, and large holes. **Right**: The image upscaled 4X by Magnific AI.

| | SDXL Inpainting | | | Ps Generative Fill | | | Magnific AI |
|---|---|---|---|---|---|---|---|
| Edit Region Ratio | 0 - 15% | 15 - 30% | 30 - 60% | 0 - 15% | 15 - 30% | 30 - 60% | 100% |
| Random Chance | 7.69% | 7.69% | 7.69% | 7.69% | 7.69% | 7.69% | 33.33% |
| Original Image | 90.96% | 90.96% | 90.96% | 90.96% | 90.96% | 90.96% | 93.33% |
| Post-Editing | 64.96% | 61.56% | 55.62% | 88.21% | 85.44% | 71.91% | 70.00% |

Table 4: Comparison of post-editing detection accuracy across different AI models. We use the best performing origin attributor in Table 1 for evaluation, which is EfficientFormer trained with text prompts. Accuracy declines at a similar rate after modifying the image using SDXL Inpainting [43] and Photoshop (Ps) Generative Fill [42].

1, EfficientFormer trained with text prompts, on these post-edited images. According to Tab. 4, we observed a monotonic decrease in accuracy with respect to the modified area of the images. Notably, SDXL Inpainting resulted in greater accuracy loss compared to Ps GenFill for the same images and masks. We hypothesize this disparity arises because the SDXL Inpainting model closely relates to the SDXL text-to-image (T2I) model included in our training generator pool, potentially skewing edited images towards an SDXL-like appearance, whereas Ps GenFill does not closely resemble any generator in our training set. This observation is validated in the corresponding confusion matrix, which we have shared in the supplemental materials. For texture enhancements via Magnific AI [33], budget constraints limited our examination to 10 examples from each of the three generators: DALL-E 3, Midjourney 6, and SDXL Turbo. This limitation set a basic random chance of classification at 33.33%. This analysis, reflected in the last column of Tab. 4, shows approximately 23% degradation, despite editing all pixels in the images. Despite the noted performance reductions, the accuracy for all post-edited images remains significantly above random chance, establishing a strong baseline for the task of post-editing image attribution.

**Image Composition Pattern.** Beyond stylistic differences, we hypothesize that various generators might create images with unique composition patterns or layouts from the same text prompt. For instance, given identical prompts, some generators may depict humans in portrait-style photos, while others may place humans further from the camera, treating them as elements within the larger scene. These variations could stem from each generator's learning with its distinctively 'curated' training data distribution and proprietary prompt augmentation techniques, features that are often integral to commercial models like DALL-E [4, 45] and Midjourney [38]. To test our hypothesis, we analyze 100 images generated from the same prompt for each generator. We employ Grounded SAM [47] to compute segmentation masks, serving as a proxy for layout representation. For instance, as depicted in Fig. 11, by averaging the segmentation masks for 'person' and 'corgi' across 100 images from each generator, created from the prompt 'a couple, a daughter, and a corgi walking,' we visualize the distribution of image composition. This reveals unique layout patterns among the generators, supporting our hypothesis.

Given the noticeable variations in the layout of generated images for a specific prompt, we further investigate whether a classifier can learn to attribute images based solely on their composition. To this end, we segment 111 semantic classes using Grounded SAM [47] and then train EfficientFormer [28] on the segmentation maps with their input prompts by concatenating their respective embeddings.

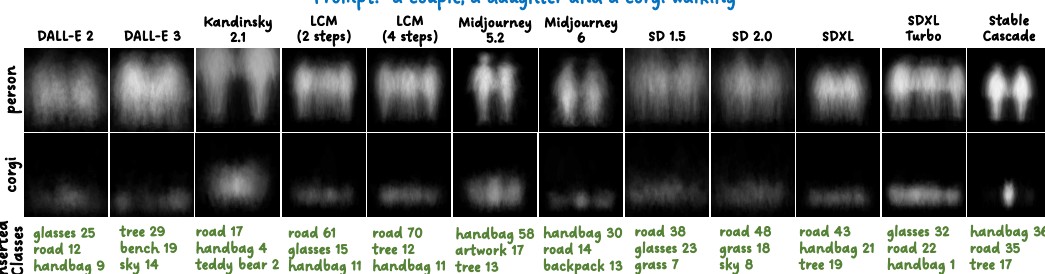

Figure 11: Image composition analysis for a given prompt. We show the averaged segmentation masks for the 'person' and 'corgi' classes. Some generators put objects at specific locations. We also list the top inserted classes and how many images (out of 100) with these classes.

This approach enables the classifier to achieve an accuracy of 17.66%, despite relying on such a coarse representation. Remarkably, this accuracy is more than twice that expected by random chance (7.69%), suggesting that distinct patterns in layout generation do indeed exist across these generators.

## D.2 Color Analysis

In addition to studying image style and image composition pattern, we examine whether different generators produce images with distinct color schemes. We use 100 images generated from a set of fixed prompts for our analysis. In Fig. 12, we visualize the density distribution of pixel values in each RGB color channel. We discover that Kandinsky 2.1 [46], Midjourney 5.2 [38], and Stable Cascade [41] often generate images with a wider range of pixel intensity values. In Fig. 13, we observe that these three generators often create images with glow and shadow effects, which can lead to higher and lower intensities.

## D.3 Comparison of Frozen vs. Fine-tuned CLIP/ DINOv2 Backbone

In Sec. 3.1 of the main paper, we evaluated the accuracy of a frozen CLIP [44] backbone connected with a linear probe and MLP, and a frozen DINOv2 [40] backbone with a similar configuration. In this section, we compare using a frozen and fine-tuned backbone for the CLIP and DINOv2 linear probes. Table 5 indicates that a CLIP backbone provides slightly better performance than a DINOv2 backbone when the backbone is frozen. However, the reverse holds true when the backbone is fine-tuned.

|  | CLIP + LP | | DINOv2 + LP | |
| --- | --- | --- | --- | --- |
| Backbone | Frozen | Fine-tuned | Frozen | Fine-tuned |
| Accuracy | 70.15% | 95.31% | 67.68% | 96.67% |
| Precision | 69.95% | 95.51% | 67.36% | 96.71% |
| Recall | 70.15% | 95.32% | 67.68% | 96.67% |
| F1 | 70.00% | 95.34% | 67.45% | 96.67% |

Table 5: Quantitative comparison of using a frozen or fine-tuned backbone to train CLIP [44] and DINOv2 [40] linear probes. CLIP achieves higher accuracy than DINOv2 when the backbone is frozen, but the opposite is true when the backbone is fine-tuned.

## D.4 Image Resolutions

The default EfficientFormer [28] takes inputs of size $224 \times 224$. We examine the performance of using five additional image resolutions between $128 \times 128$ and $1024 \times 1024$ for origin attribution. As illustrated on the left side of Fig. 14, accuracy tends to increase as image resolution increases.

## D.5 Cropped Image Patches

Our previous experiments use most, if not all, image pixels for the origin attribution task. We also explore the opposite: how few pixels are necessary to achieve good performance? Inspired by [8, 66], we crop a single patch of each image and then train EfficientFormer [28] on these patches instead of the full-sized images. Specifically, we first resize each original image to have a shorter edge of size 512, then center crop the image to create a patch of size $k \times k$, and finally resize the patch to $224 \times 224$. We utilized $k = [2, 4, 8, 16, 32, 64, 128, 256]$ and resized images using bicubic interpolation. On the right side of Fig. 14, we see that accuracy increases with image patch size. Remarkably, even training an origin attributor on $2 \times 2$ patches can lead to 22.29% accuracy, which is well above the random chance accuracy of 7.69%.

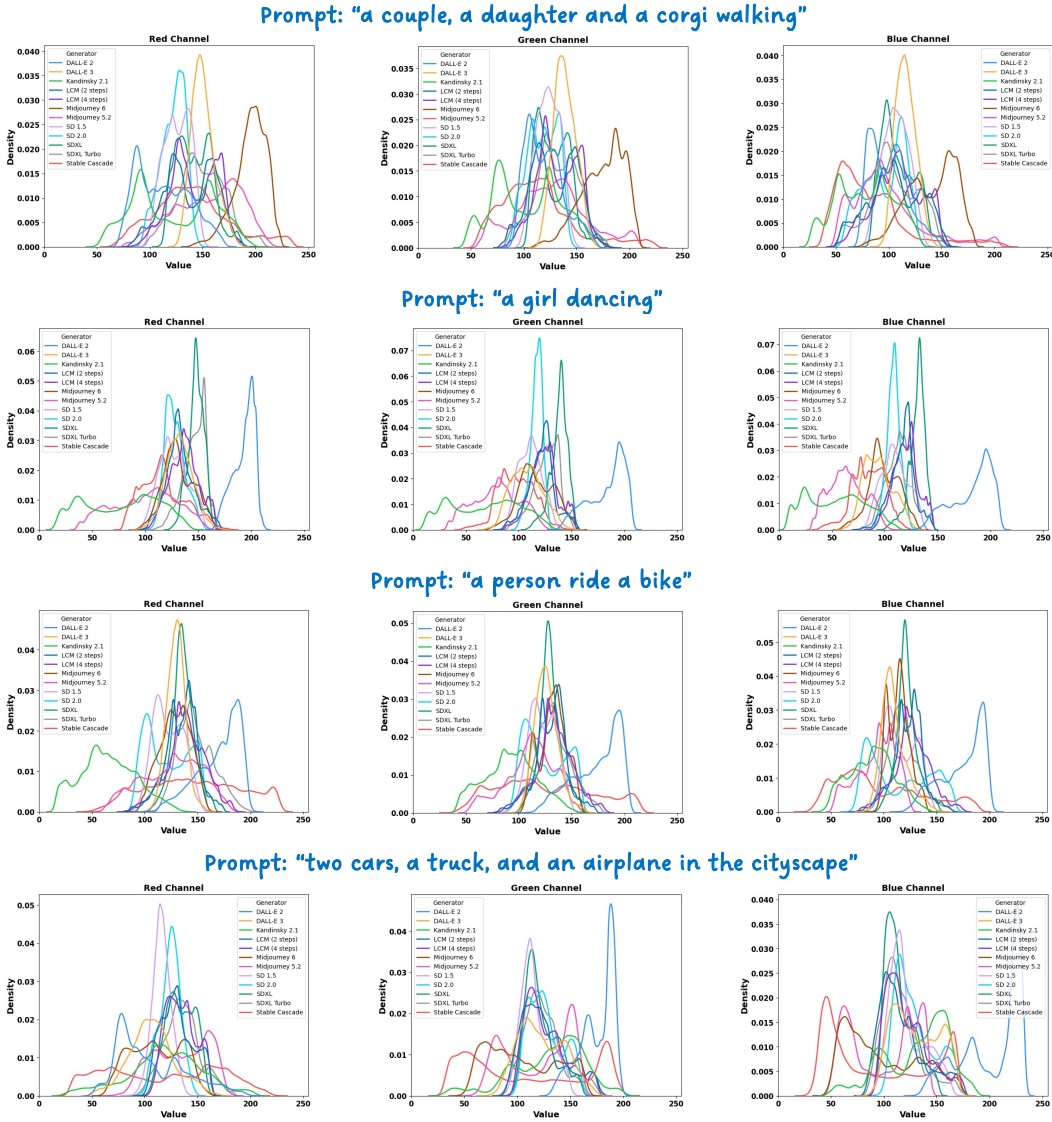

Figure 12: Density distribution of pixel values in RGB color channels after averaging 100 images for each prompt and generator. Kandinsky 2.1 [46], Midjourney 5.2 [38], and Stable Cascade [41] tend to create images covering a wider range of pixel intensities.

## D.6  Potential Application of Model Stealing

It's important to note that our research might facilitate 'model stealing,' or the reverse engineering of a model's architecture. As an initial experiment, we projected 20 images generated from each of the four most recent non-open-source models—'Adobe Firefly Image 3' [1], 'SD 3' [58], 'SD 3 Turbo' [58], and 'Meta AI Imagine' [37]—into the t-SNE feature embedding space of our pretrained origin attributor. As illustrated in Fig. 15, we observe that images from 'Adobe Firefly Image 3' appear similar to those from 'Midjourney 5.2' and real images. Meanwhile, 'SD 3' and 'SD 3 Turbo' are closer to 'Stable Cascade' and 'Midjourney 6', and 'Meta AI Imagine' largely overlaps with 'DALL-E 3'. This comparative analysis could lay the groundwork for inferring the architectures of non-open-source models based on those already known.

## E  Grad-CAM Visualizations

Figure 16 showcases the Grad-CAM [20, 53] heatmaps for origin attributors trained on various image types, including the original RGB images, images after high-frequency perturbations, and mid-level representations. We observe that the origin attributors trained on RGB images and images after high-frequency perturbations tend to pay attention to smooth image regions, such as the sky or ground. Nonetheless, even though the attributors focus on varied image regions, it remains difficult to explain how they make their decisions for each image.

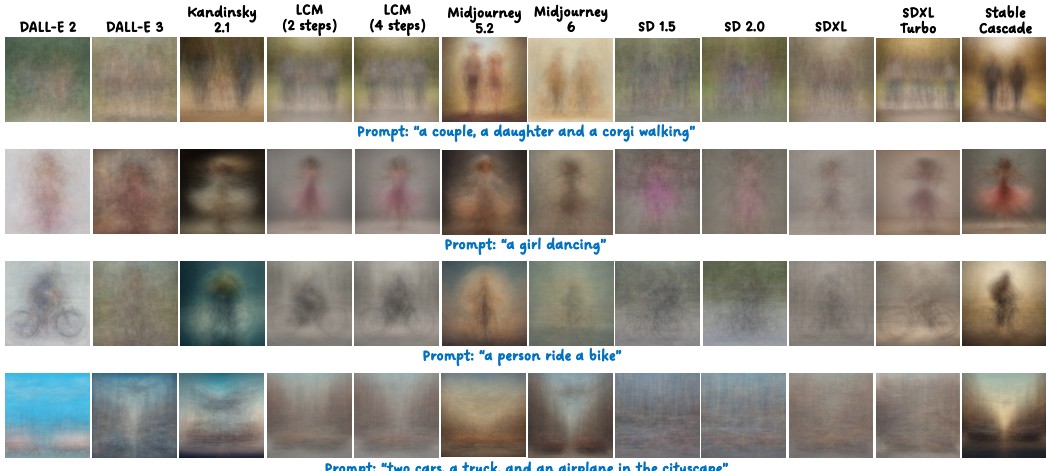

Figure 13: Visualization of 100 images averaged together for each prompt and generator. Consistent with our observations in Fig. 12, we see that Kandinsky 2.1 [46], Midjourney 5.2 [38], and Stable Cascade [41] often produce images with glow and shadow effects.

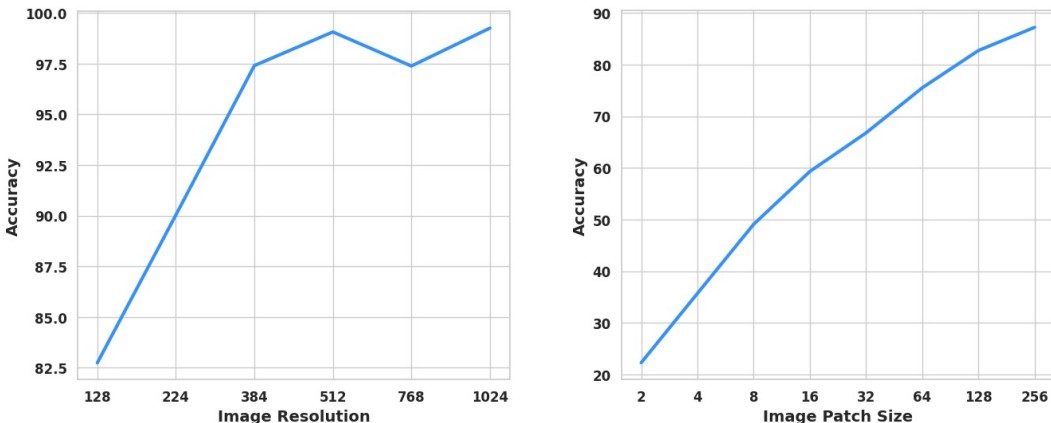

Figure 14: **Left:** Accuracy of our EfficientFormer [28] image attributor across six image resolutions on the 13-way classification task. In general, accuracy increases as image resolution increases. **Right:** Accuracy of EfficientFormer across eight image patch sizes. Interestingly, using $2 \times 2$ image patches can achieve 22.29% accuracy, whereas the probability of randomly guessing the correct generator is $\frac{1}{13}$, corresponding to 7.69%.

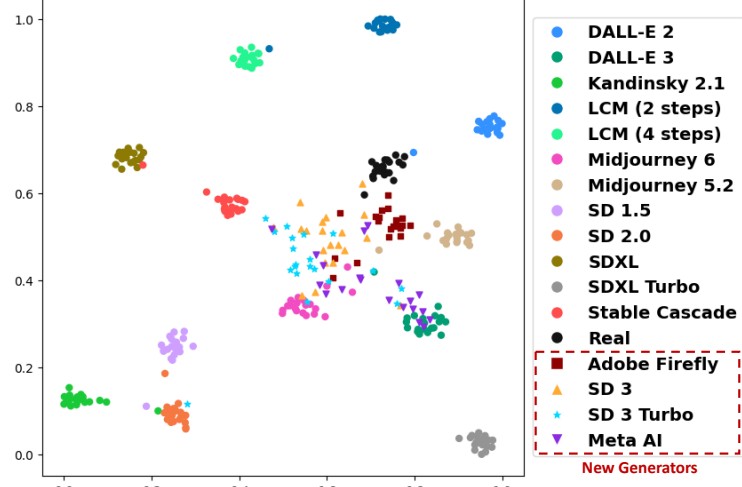

Figure 15: A t-SNE visualization of 4 unseen new generators in the feature space of our pretrained origin attributor.

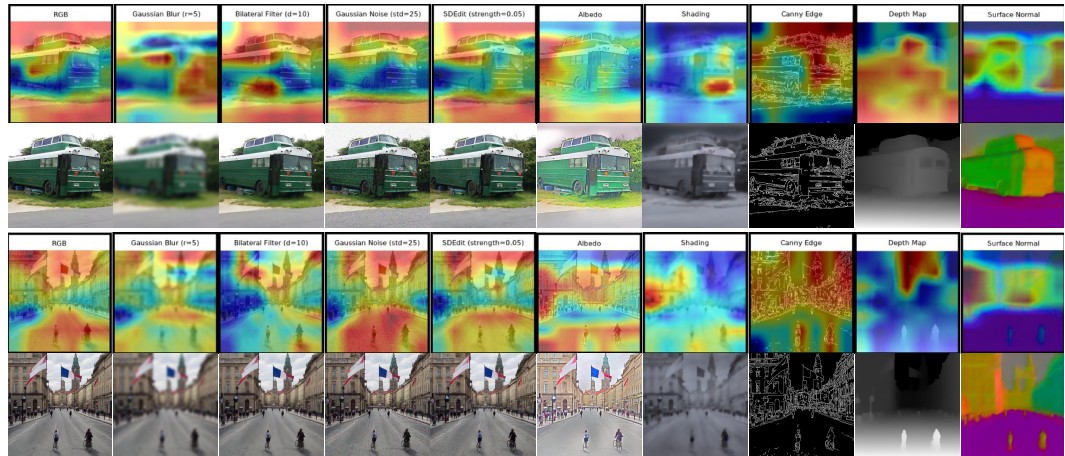

Figure 16: Grad-CAM [20, 53] visualizations for image attributors trained on each image type, where each column represents a distinct attributor. The first and third rows illustrate the Grad-CAM heatmaps overlaid on the input images. The second and fourth rows show the input images without Grad-CAM. The first example on the top is based on a real image from MS-COCO [29], while the second example on the bottom is based on a fake image generated by SDXL Turbo [51]. We notice that the attributors trained on RGB images and images after high-frequency perturbations often focus on relatively smooth image regions, such as the sky or ground.

## F   Broader Impacts

We acknowledge that text-to-image diffusion models pretrained on large-scale, uncurated web data may produce biases and errors. Additionally, we use text prompts that are based on captions of MS-COCO [30] images, which may generate images of people.

