# OpenReview forum: "Detecting Origin Attribution for Text-to-Image Diffusion Models in RGB and Beyond"
_NeurIPS.cc/2024/Workshop/SafeGenAi — SafeGenAi Poster_

### Official Review · Reviewer_QmJ1 · 2024-10-08
**This paper analyzes the detection and attribution of images generated by modern text-to-image (T2I) diffusion models; however, it could be further improved in terms of methodological clarity and result interpretation.**

**Rating:** 5
**Confidence:** 4

**Review:**

1. The introduction provides a broad overview of T2I models, but it could be enhanced by systematically comparing existing attribution techniques, particularly for diffusion models. A more explicit discussion on how this paper’s contributions surpass previous GAN-based image attribution studies would help highlight the paper’s novelty and innovative approach.

2. The authors explore different hyperparameters (e.g., checkpoint variations, scheduler type, seed, and sampling steps) but do not thoroughly explain the rationale behind each choice.

3. While Table 2 presents the accuracy for detecting various hyperparameter configurations, it lacks detailed analysis of the results. For example, why does seed variation achieve nearly 100% accuracy?

4. There are several formatting issues with the references. For example, references 6-8 lack publication details.

5. There are several instances where quotation marks are incorrectly formatted.

---

### Official Review · Reviewer_EPHK · 2024-10-10
**The contribution is not well presented**

**Rating:** 4
**Confidence:** 4

**Review:**

Strength:
This topic is important for the theme of this workshop.

Weakness:
1. The contribution of this paper is not very well presented. In Section 3, it is not clear how the original attribute in RGB is detected.
2. The study extensively analyzes the effects of high-frequency perturbations, yet the practical impact is limited. Real-world manipulations may involve more complex transformations that aren't fully addressed, such as style transfer or aggressive upscaling.

---

### Official Review · Reviewer_FyfC · 2024-10-11
**Review of 23**

**Rating:** 9
**Confidence:** 5

**Review:**

This paper proposes a framework that can identify which T2I generator created a particular image and also investigate key hyperparameters used during the inference stage. By leveraging both high-frequency and mid-level image representations, the proposed model achieves high accuracy in distinguishing images generated by 12 different diffusion models. And beyond RGB, mid-level representations are also explored, enhancing attribution performance.

### Strengths

1. This paper is well structured and clearly written. The inclusion of both RGB and mid-level representations offers a holistic view of how different visual traces can be leveraged for attribution. The exploration beyond RGB is interesting and provides new perspectives on how image attributors can distinguish between synthetic and real images.
2. The investigation into hyperparameter variations is also good and provides valuable insights.

### Weaknesses

1. The paper relies heavily on empirical results but does not provide a detailed theoretical explanation for why mid-level representations perform well for attribution. It would be better to provide a deeper theoretical discussion to strengthen the contribution of this work